# Pre-Charge Pressure Estimation of a Hydraulic Accumulator Using Surface Temperature Measurements

Magnus F. Asmussen [1], Jesper Liniger [2,*], Nariman Sepehri [3] and Henrik C. Pedersen [4]

1    Nel Hydrogen, DK-7400 Herning, Denmark
2    AAU Energy, Aalborg University, DK-6700 Esbjerg, Denmark
3    Department of Mechanical Engineering, University of Manitoba, Winnipeg, MB R3T 5V6, Canada
4    AAU Energy, Aalborg University, DK-9220 Aalborg East, Denmark
*    Correspondence: jel@energy.aau.dk; Tel.: +45-93562090

**Abstract:** Pitch systems form an essential part of today's wind turbines; they are used for power regulation and serve as part of a turbine's safety system. Hydraulic pitch systems include hydraulic accumulators, which comprise a crucial part of the safety system, as they are used to store energy for emergency shutdowns. However, accumulators may be subject to gas leakage, which is the primary failure mode. Gas leakage affects the performance of the accumulator and, in extreme cases, compromises the safety function of the pitch system. This paper deals with the development and experimental validation of an algorithm to detect gas leakage in piston-type accumulators. The innovation of the algorithm is the ability to generate estimates of the remaining amount of gas while solving the drift problem evidenced in previous research. Additionally, this method enables the ability to isolate gas leakage to a single accumulator out of a bank of accumulators. The approach is based on a State Augmented Extended Kalman Filter (SAEKF), which utilizes an extended thermal model of the accumulator, as well as temperature measurements along the accumulator surface to estimate the remaining gas in the accumulator. The method is experimentally validated and addresses the drift problem in estimating the gas leakage evidenced from previous research. Additionally, the method can identify and isolate gas leakage to a single accumulator from a bank of accumulators.

**Keywords:** fault detection and diagnosis; fluid power; accumulator

## 1. Introduction

Hydraulic piston accumulators are utilized in a wide range of applications; one example is the pitch control system for large wind turbines, which is used to control the pitch of the blades. In the pitch system, the piston accumulators are primarily used for the safety system, where the hydraulic energy stored in the piston accumulators is used during emergency shutdowns. The piston accumulators are also used for the supply system, where they are used for increased efficiency, decreased pressure spikes, and to supplement the pump in periods with high flow demands. Therefore, the accumulators' function makes them a crucial part of the system. According to Carroll et al. [1], accumulators are responsible for 10.7% of the failures in the pitch control system in wind turbines, making it the third-most-common failure. The main failure mode of a gas-charged piston accumulator is the loss of pre-charge pressure, i.e., gas leakage, reducing the energy-storing ability of the accumulator. Direct measurement of the accumulator piston position to detect gas leakage is costly [2] and further introduces extra leakage paths. Therefore, to increase the overall reliability and decrease the service costs, enhanced methods for detecting the reduced pre-charge pressure of piston accumulators should be introduced. Different methods have been addressed in the review by Asmussen et al. [3]. The following presents methods based on that review with the addition of more recent research.

Elorza et al. considered a signal-based method for detecting several faults in a hydraulic pitch system [4]. Accumulator gas leakage was also considered, and a simulation

study showed the ability to detect a 50% drop in pre-charge pressure. This is a significant change in pre-charge pressure, which questions the ability of the method to detect the fault before developing into a failure.

Nisters et al. presented an accumulator gas leakage detection using only a fluid pressure sensor [5]. They showed that a significant pressure decay occurs when the piston reaches the end stop, where fluid is completely expelled from the accumulator. Therefore, the method relies on emptying the accumulator to access the pre-charge pressure of the gas. Similar methods for pitch systems in wind turbines are presented in the patents by Nielsen et al. [6] and Minami et al. [7].

Liniger et al. [8] described a method for detecting the changes in the pre-charge pressure of an accumulator used in wind turbines. The method is based on wavelet analysis and uses a measurement of the fluid pressure close to the accumulator. The RMS value of the detail coefficient, corresponding to a frequency range of 0.39–0.78 Hz, in which a known excitation frequency for wind turbines is located, is used to quantify changes in the pre-charge pressure. Through simulations, it was shown that the method was able to distinguish pre-charge pressures of 180 bar, 100 bar, and 50 bar at an ambient temperature range of 22 to 60 °C. The method was further validated experimentally, to distinguish 100 bar, 75 bar, and 50 bar pre-charge pressure levels. The detectable levels suffice for evaluating if the accumulators need gas replenishment or not.

In [9], Liniger et al. described a method for detecting gas leakage using an Extended-Kalman-Filter (EKF)-based algorithm. The method utilized fluid pressure measurements and ambient temperature measurements to estimate the accumulator's pre-charge pressure. Experiments showed that the pre-charge pressure could be estimated when using measurements of the input flow of the accumulator. However, the estimate of the pre-charge pressure was drifting when an estimated input flow was used. Flow measurements are generally not present in hydraulic pitch systems; thus, the method may not be applied to monitor the pre-charge pressure continuously. The pre-charge pressure was estimated during the charging of the accumulator with an estimation error within $\pm 2$ bar.

Helwig et al. [10] detected accumulator gas leakage, among other faults, by multivariate statistics based on multiple signals to extract features related to faults. The method was tested experimentally when subjected to both fixed and random loading cycles and could distinguish between 90 bar, 100 bar, 110 bar, and 115 bar pre-charge pressures for fixed working cycles. However, the method showed decreased performance when detecting the change in pre-charge pressure for random load cycles. Due to the random wind speed affecting the blades, accumulators in wind turbines are exposed to random loading cycles. Several other studies have considered the development of fault-detection algorithms using machine learning on the same set of experimental data; yet, the robustness to random loads when operating remains a challenge [11–13].

Haas and Pichler used a Support Vector Machine (SVM), trained by a simulation model, to detect and isolate pump leakage and accumulator gas leakage faults in a simple loading circuit [14]. The method was experimentally validated in a pure loading cycle. However, the method's performance in the case of varying load flow is not clear.

Jakobsson et al. investigated fault detection in hydraulic rock-drilling equipment [15]. They showed that accumulator gas leakage could be detected by estimating the pressure gradient in a known load cycle. However, the load cycle is unknown for a pitch system, and the method is not directly applicable.

Sorensen et al. [16] described a method using a bank of EKFs to detect changes in the pre-charge pressure of a piston accumulator. The residuals of four EKFs with different assumed pre-charge pressures were analyzed using a multi-model adaptive estimation scheme to evaluate the most-likely pre-charge level. Through experiments, the method was shown to be capable of isolating 140 bar, 110 bar, 80 bar, and 50 bar from each other. However, the method relied on a flow measurement in the accumulator, which is generally not present in standard pitch systems.

A common challenge regarding several of the methods above is that they use measurements typically unavailable in hydraulic pitch control systems, e.g., flow or vibration measurements. In addition, using estimates instead of measurements, the methods experience problems with drifting. Another challenge is that the detection level is coarse, in the range of 25–30 bar [4,8,9,16]. More detailed detection levels further the ability to determine the remaining useful life, thus enhancing the planning of maintenance tasks.

Therefore, one of the ideas in this paper is the possibility of estimating the pre-charge directly by using the gas equation of state and the piston position. However, as described above, using a position sensor is not desirable as it is costly and not readily available in hydraulic pitch systems. The work presented in this paper was thus motivated by utilizing the relationship between the accumulator surface temperature and the piston position under the excitation of the accumulator pressure, as given experimentally in Figure 1. Figure 1 shows the mean surface temperature of a 5 min pressure charging sequence for three different pre-charge pressures. The fluid pressure is initially stepped from atmospheric to 180 bar and subsequently kept constant. The vertical black bar indicates the measured final piston position. The surface temperatures are clearly elevated by the gas temperature increase due to the gas compression. The experiment suggests that the surface temperature can be used as an indicator of the piston position. Yet, it should be noticed that accumulators are in general operated under many different load scenarios where the relation between piston position and surface temperature is not as clear as given in this illustrative experiment; hence, the detection algorithm developed in this paper utilizes more information than just surface temperatures.

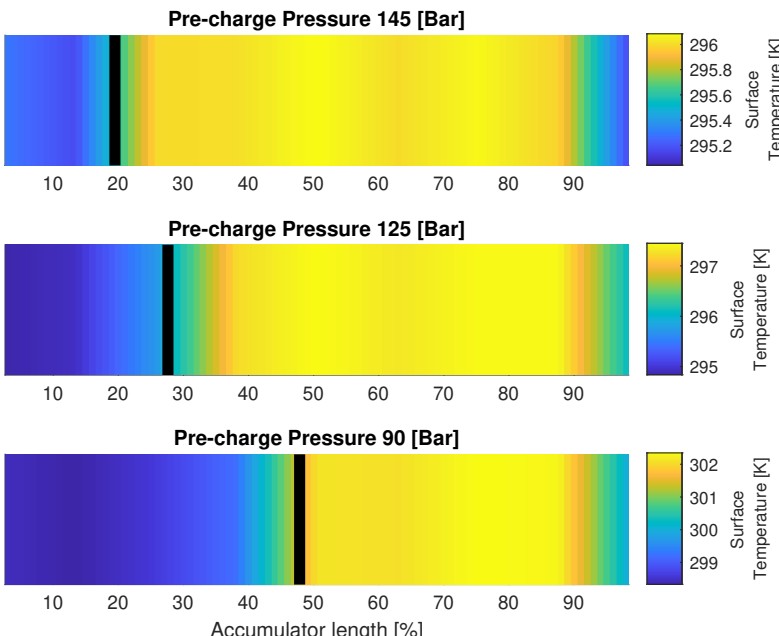

**Figure 1.** Mean surface temperatures for different pre-charge pressures. The fluid inlet is located on the left-hand side of the plot (also seen in Figure 2), which also denotes the initial piston position. The vertical black bar shows the final piston position. The surface temperature values are interpolated from eight evenly distributed sensors along the length of the accumulator.

In previous model-based approaches [9,16,17], the accumulator is treated as a lumped mass, and the heat transfer to the surroundings is described with a thermal time constant. This approximation was described by Rotthauser [18]. Hansen and Rasmussen [17] suggested that the thermal model may be extended for a more accurate model. Hence, to improve the methods that have been discussed, it is suggested to extend the thermal model of the accumulator, inspired by the models presented by Pfeffer [19], where three thermal models for a carbon-fiber-reinforced plastic housing were developed. Even though this

paper deals with steel accumulators, the approach is still valid for other accumulator types. By extending the thermal model to include the temperature of surface elements and end caps, it is expected that the pre-charge pressure estimation can be improved. Furthermore, the potential drifting seen [9] when using estimated flows instead of measurements can be removed as the presented method does not rely on flow measurements or estimated flow.

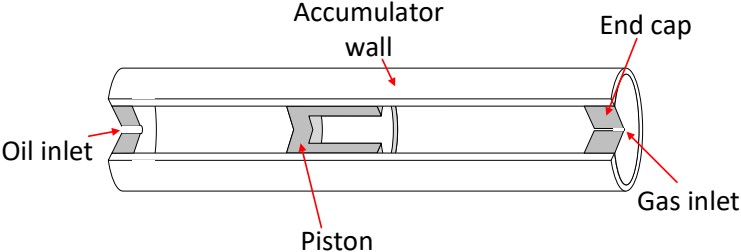

**Figure 2.** Schematic of the accumulator showing internal parts.

In this paper, a State Augmented Extended Kalman Filter (SAEKF) with an extended thermal model is proposed to estimate the pre-charge pressure of a hydraulic piston accumulator. Compared to the previously published models, the thermal model of the SAEKF is extended, where the temperature of the accumulator surface is included to increase the accuracy of the thermal model. The presented method uses temperature measurements along the accumulator surface, the ambient temperature, the oil temperature, and the oil pressure in the detection algorithm. By using surface temperatures, the proposed method does not rely on the estimation of the input fluid flow, which was seen to cause estimation drift issues. Accumulators are often arranged in a bank of multiple accumulators. Since the input oil flow is not needed for detection, the proposed method can isolate faults to a single accumulator out of a bank of accumulators, rendering the maintenance procedure more effective.

## 2. Accumulator Model

The accumulator model is divided into three parts; a mechanical model, a gas model, and a thermal model. All three models will be described in the following, starting with the mechanical model. The gas leakage is not explicitly modeled, as it either occurs in discrete events caused by seal blow-by or slowly develops over weeks or even months. Hence, it is suitable to consider the amount of gas constant during the time frame of detection considered in this paper. The model parameter values are collected in Table 1.

### 2.1. Mechanical Model

The piston movement of the accumulator is modeled by Newton's second law of motion as:

$$\ddot{x}_p\, m_p = (p_o - p_g)A_p - F_{fric} \tag{1}$$

where $x_p$ is the piston position, $p_o$ is the oil pressure, $p_g$ is the gas pressure, $A_p$ is the piston area, and $m_p$ is the mass of the piston. A simple friction model is used and $F_{fric}$ is given as:

$$F_{fric} = B_{fric}\dot{x}_p + C_{fric}\, sign(\dot{x}_p) \tag{2}$$

where $B_{fric}$ is the viscous friction constant and $C_{fric}$ is the Coulomb friction constant.

### 2.2. Gas Model

The gas inside the accumulator is nitrogen, and it is modeled by the Benedict–Webb–Rubin (BWR) equation of state [21], which is given by:

$$p_g = \frac{RT_g}{v} + \frac{B_0 RT_g - A_0 - \frac{C_0}{T_g^2}}{v^2} + \frac{bRT_g - a}{v^3} + \frac{a\alpha}{v^6} + \frac{c\left(1 + \frac{\gamma}{v^2}\right)e^{-\frac{\gamma}{v^2}}}{v^3 T_g^2} \tag{3}$$

where $T_g$ is the gas temperature, $v$ is the specific molar volume, and $(A_0, B_0, C_0, a, b, c, \alpha, \gamma, R)$ are the gas constants for nitrogen [21]. The specific molar volume is determined as a function of the piston positions such that:

**Table 1.** Parameter values used in the model. The values for $\tau, \alpha_a, \alpha_o$, and $\alpha_g$ are not stated, but left as tuning parameters and evaluated during model validation [20,21].

| Notation | Description | Value | Unit |
|---|---|---|---|
| $m_p$ | Piston mass | 3.97 | (kg) |
| $B$ | Viscous friction coefficient | $5 \times 10^3$ | $\left(\frac{\text{N·s}}{\text{m}}\right)$ |
| $C$ | Coulomb friction coefficient | $2 \times 10^3$ | (N) |
| $R$ | Gas constant | 8.31 | (N) |
| $A_0$ | BWR constant | 0.11 | () |
| $B_0$ | BWR constant | $4.07 \times 10^{-5}$ | () |
| $C_0$ | BWR constant | 816.58 | () |
| $a$ | BWR constant | $2.54 \times 10^{-6}$ | () |
| $b$ | BWR constant | $2.33 \times 10^{-9}$ | () |
| $c$ | BWR constant | $7.38 \times 10^{-2}$ | () |
| $\alpha$ | BWR constant | $1.27 \times 10^{-13}$ | () |
| $\gamma$ | BWR constant | $5.3 \times 10^{-9}$ | () |
| $c_{N2}$ | Specific heat capacity of nitrogen | 1040 | $\left(\frac{\text{J}}{\text{kg·K}}\right)$ |
| $M_{N2}$ | Molar mass of nitrogen | $2.8 \times 10^{-2}$ | $\left(\frac{\text{kg}}{\text{mol}}\right)$ |
| $V_{a,0}$ | Non-displaceable accumulator volume | $1.4 \times 10^{-3}$ | (m$^3$) |
| $m_e$ | End-cap mass | 28.45 | (kg) |
| $c_{steel}$ | Specific heat capacity of steel | 490 | $\left(\frac{\text{J}}{\text{kg·K}}\right)$ |
| $\alpha_a$ | Heat transfer coefficient (air to steel) | - | $\left(\frac{\text{W}}{\text{m}^2\text{·K}}\right)$ |
| $\alpha_o$ | Heat transfer coefficient (oil to steel) | - | $\left(\frac{\text{W}}{\text{m}^2\text{·K}}\right)$ |
| $\alpha_g$ | Heat transfer coefficient (gas to steel) | - | $\left(\frac{\text{W}}{\text{m}^2\text{·K}}\right)$ |
| $A_{eg}$ | Area between end-cap and gas | $2.54 \times 10^{-2}$ | (m$^2$) |
| $A_{ea}$ | Area between end-cap and air | $9.18 \times 10^{-2}$ | (m$^2$) |
| $A_{pg}$ | Area between piston and gas | $2.71 \times 10^{-2}$ | (m$^2$) |
| $A_p$ | Piston area | $2.54 \times 10^{-2}$ | (m$^2$) |
| $r_{is}$ | Inner radius of accumulator | $9 \times 10^{-2}$ | (m) |
| $r_{os}$ | Outer radius of accumulator | $11 \times 10^{-2}$ | (m) |
| $n$ | Number of wall elements | 8 | (-) |
| $d_x$ | Length of wall element | 0.123 | (m) |
| $l_p$ | Piston length | 0.11 | (m) |
| $l_a$ | Accumulator length | 0.983 | (m) |
| $\rho$ | Density of steel | 7800 | $\left(\frac{\text{kg}}{\text{m}^3}\right)$ |

$$v = \frac{A_p \left( l_a - x_p \right) + V_{a,0}}{n_{N2}} \tag{4}$$

where $l_a$ is the maximum stroke of the piston, $V_{a,0}$ is the non-displaceable volume, and $n_{N2}$ is the amount of nitrogen contained in the accumulator in moles.

The temperature of the gas is found from the first law of thermodynamics, where the gas is considered a closed system with a uniform temperature distribution. Following the deduction in [22], the change in temperature of the gas can be derived as:

$$\dot{T}_g = -\frac{\dot{Q}_s}{c_{N2}\, n_{N2}\, M_{N2}} - \frac{\dot{v} T_g}{c_{N2}\, M_{N2}} \left( \frac{\partial p_g}{\partial T_g} \right) \tag{5}$$

where $\dot{Q}_s$ is the heat transfer from the gas to the interfacing thermal elements, as seen in Equation (24), $c_{N2}$ is the specific heat capacity of nitrogen, $m_{gas}$ is the mass of the gas, and $M_{N2}$ is the molar mass of nitrogen. By assuming a constant amount of nitrogen, the time derivative of $v$ is given as:

$$\dot{v} = -\frac{A_p \dot{x}_p}{n_{N2}} \tag{6}$$

### 2.3. Thermal Model

For the thermal model, the accumulator is divided into three parts: the piston, the end cap in the end with the gas inlet, and the accumulator wall. The different parts are shown in Figure 2. The thermal model is based on Model 2 in Pfeffer et al. [19].

The temperature of the end cap towards the oil inlet is assumed to be equal to the oil temperature as it is in contact with the oil during operation. In the following, the thermal models of the different parts are elaborated, starting with the end cap at the gas inlet.

#### 2.3.1. End Cap

The gas inlet end cap is modeled as a lumped thermal body with a uniform temperature and solving with respect to the temperature gradient [20]:

$$\dot{T}_e = \frac{\dot{Q}_{ge} + \dot{Q}_{ae}}{m_e c_{steel}} \tag{7}$$

where $T_e$ is the temperature of the end cap, $m_e$ is the mass of the end cap, $c_{steel}$ is the specific heat capacity of steel, and $\dot{Q}_{ge}$ and $\dot{Q}_{ae}$ are the heat flows from the gas and surroundings to the end cap, respectively. $\dot{Q}_{ge}$ and $\dot{Q}_{ae}$ are described as convective heat transfers and are found in Equations (8) and (9) [20].

$$\dot{Q}_{ge} = \alpha_g A_{eg} \left( T_g - T_e \right) \tag{8}$$

$$\dot{Q}_{ae} = \alpha_a A_{ea} \left( T_a - T_e \right) \tag{9}$$

where $T_a$ is the ambient temperature, $\alpha_a$ and $\alpha_g$ are convective heat transfer coefficients, and $A_{ea}$ and $A_{eg}$ are the surface areas of the end towards the surroundings and the gas, respectively.

#### 2.3.2. Accumulator Piston

The piston is modeled in the same way as the end cap, thus yielding:

$$\dot{T}_p = \frac{\dot{Q}_{gp} + \dot{Q}_{op}}{m_p c_{steel}} \tag{10}$$

$$\dot{Q}_{gp} = \alpha_g A_{pg} \left( T_g - T_p \right) \tag{11}$$

$$\dot{Q}_{op} = \alpha_o A_p \left( T_o - T_p \right) \tag{12}$$

where $T_p$ is the temperature of the accumulator piston, $\alpha_g$ and $\alpha_o$ are convective heat transfer coefficients, and $A_{pg}$ and $A_p$ are the surface areas of the piston towards the gas and the oil, respectively.

### 2.3.3. Accumulator Wall

The accumulator is divided into eight wall elements to model the temperature distribution along the wall, as shown in Figure 3.

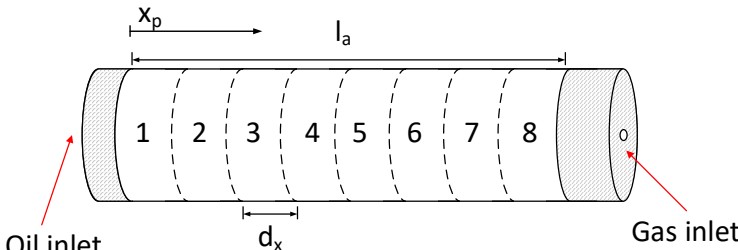

**Figure 3.** Accumulator wall schematic divided into eight elements including the piston position notation.

The temperature distribution is thus considered uniform for each wall element. Eight elements are chosen as a compromise between model complexity and model accuracy.

The heat flows between the accumulator elements are described as conductive heat flows. The energy balance for each wall element may thus be divided into a convective part and a conductive part such that:

$$\dot{T}_w^{[i]} = \dot{T}_{convective}^{[i]} + \dot{T}_{conductive}^{[i]} \tag{13}$$

where $i$ denotes the wall element number. An illustration of a wall element is shown in Figure 4 depicting the heat flows $\dot{Q}_{gw}$, $\dot{Q}_{ow}$, and $\dot{Q}_{aw}$. The surface contact areas for gas–wall $A_{gw}$, oil–wall $A_{ow}$, and ambient–wall $A_{aw}$ interaction are also shown in Figure 4.

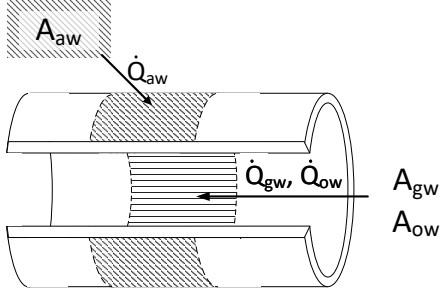

**Figure 4.** Wall element with surface areas and heat flow denotation.

The convective part for the $i$'th element can be described as:

$$\dot{T}_{convective}^{[i]} = \frac{\dot{Q}_{aw}^{[i]} + \dot{Q}_{ow}^{[i]} + \dot{Q}_{gw}^{[i]}}{\rho \, c_{steel} \left( r_{os}^2 - r_{is}^2 \right) \pi d_x} \tag{14}$$

where $r_{os}$ and $r_{is}$ are the outer and inner radius of the accumulator, respectively, and $d_x$ is the length of the wall element. The heat flows, shown in Figure 4, are then calculated as:

$$\dot{Q}_{ow}^{[i]} = \alpha_o A_{ow} \left( T_o - T_w^{[i]} \right) \xi_o^{[i]} \tag{15}$$

$$\dot{Q}_{gw}^{[i]} = \alpha_g A_{gw} \left( T_g - T_w^{[i]} \right) \xi_g^{[i]} \tag{16}$$

$$\dot{Q}_{aw}^{[i]} = \alpha_a A_{aw} \left( T_a - T_w^{[i]} \right) \tag{17}$$

where $\zeta_o$ and $\zeta_g$ describe the ratio of which the wall element is in contact with oil or gas, respectively. They are given in Equations (18) and (19). A similar approach was used in the work by Pfeffer et al. [19]. The contact areas are given as $A_{ow} = A_{gw} = d_x \, 2\pi \, r_{is}$.

$$\zeta_o^{[i]} = \begin{cases} 1 & \text{for} & x_p > i \, d_x \\ \dfrac{x_p - (i-1)d_x}{d_x} & \text{for} & i \, d_x \leq x_p \leq i \, d_x \\ 0 & \text{for} & x_p < i d_x \end{cases} \tag{18}$$

$$\zeta_g^{[i]} = \begin{cases} 0 & \text{for} & x_p + l_p > i \, d_x \\ 1 - \dfrac{x_p - l_p + i d_x}{d_x} & \text{for} & (i-1)d_x \leq x_p + l_p \leq i \, d_x \\ 1 & \text{for} & x_p + l_p < i d_x \end{cases} \tag{19}$$

where $l_p$ is the length of the accumulator piston. Note that, in the above equations, the heat transfer between the piston and the accumulator surface is neglected, as it is significantly smaller than the other heat transfers in the model.

The conductive heat flows between the wall elements using only the axial component are given as [20]:

$$\dot{T}_{conductive} = \frac{1}{\rho \, c_{steel}} \frac{\partial}{\partial x} \left( \lambda_x \frac{\partial T_w}{\partial x} \right) \tag{20}$$

The term $\frac{\partial}{\partial x} \left( \lambda_x \frac{\partial T_w}{\partial x} \right)$ is estimated using a second-order central difference equation such that:

$$\dot{T}_{conductive}^{[i]} = \frac{\lambda_x}{\rho c_{steel}} \frac{T_w^{[i-1]} - 2T_w^{[i]} + T_w^{[i+1]}}{d_x^2} \tag{21}$$

The end conditions are set to the temperature of the accumulator end caps such that:

$$\dot{T}_{conductive}^{[1]} = \frac{\lambda_x}{\rho c_{steel}} \frac{T_o - 2T_w^{[1]} + T_w^{[2]}}{d_x^2} \tag{22}$$

and

$$\dot{T}_{conductive}^{[8]} = \frac{\lambda_x}{\rho c_{steel}} \frac{T_w^{[7]} - 2T_w^{[8]} + T_e}{d_x^2} \tag{23}$$

The total heat flow from the gas to the accumulator is hence defined as:

$$\dot{Q}_s = \left( \sum_{i=1}^{8} \dot{Q}_{gw}^{[i]} \right) + \dot{Q}_{gp} + \dot{Q}_{ge} \tag{24}$$

## 3. Model Validation

The test bench consisted of a 25 L accumulator with valves controlling the flow to the accumulator. The size of the accumulator is similar to those used in hydraulic pitch control systems for multi-MW wind turbines. Figure 5 shows the test bench. The test sequence used to generate the experimental data was chosen to simulate the typical operation for an accumulator placed in the supply unit of a hydraulic pitch control system, which was also used in previous works [8,9]. Here, the pressure level is controlled between two thresholds chosen as 165 bar and 190 bar. A constant flow supply is turned on when the pressure is below the lower threshold. The supply is turned off when the pressure rises to the upper threshold pressure. The fluid flow from the accumulator is controlled in a closed loop to simulate the flow demand from a pitch system, where the flow varies due to the volatility of the pitch activity.

For the validation of the model, thermocouples were attached to the accumulator, as shown in Figure 6. For validation purposes, the piston position was also measured.

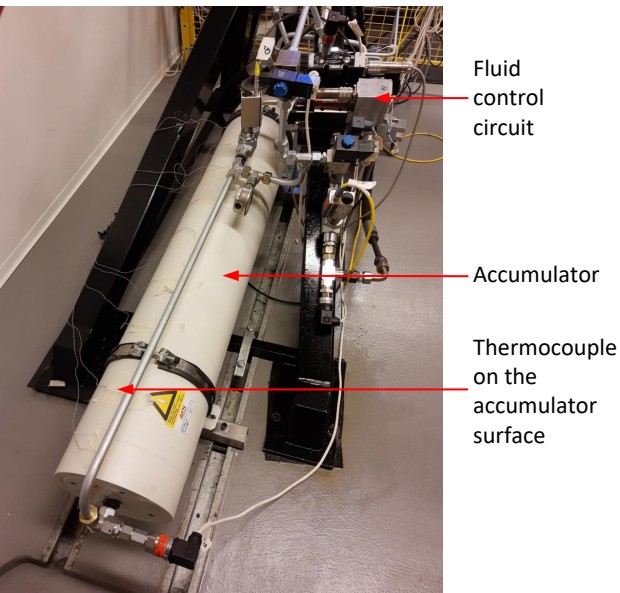

**Figure 5.** Test bench used to generate experimental data.

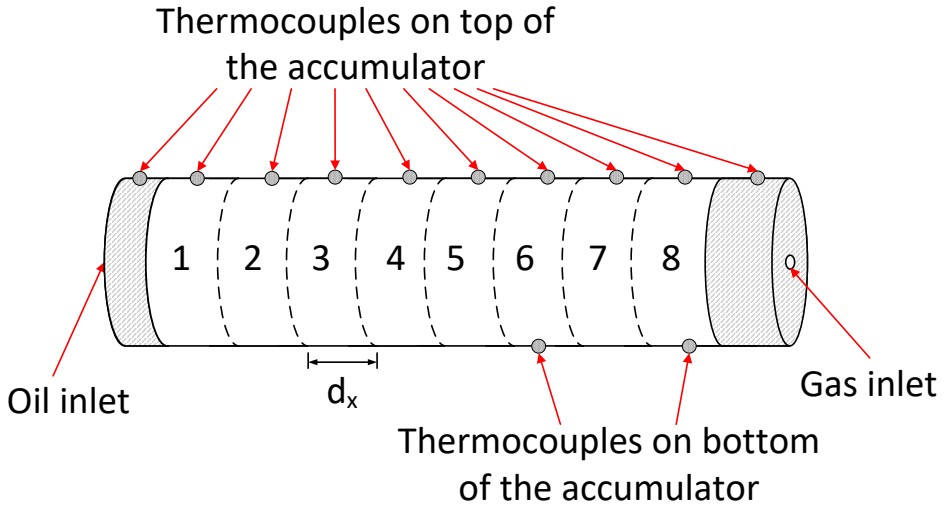

**Figure 6.** Placement of the thermocouples attached to the accumulator surface.

As shown in Equation (25), the oil pressure, ambient temperature, and oil temperature are given as the input. The model states used for validating the model are the accumulator piston position and temperatures, which are compared to the measured values; cf. Equation (25). The ambient temperature is measured in the vicinity of the accumulator. The distance to the accumulator should be chosen such that the measurement is unaffected by radiation. In practice, a distance above a couple of centimeters suffices. The oil temperature is measured at the oil inlet of the accumulator.

Figure 7 shows the simulated and measured piston position and the measured oil pressure. For the simulation, the heat transfer coefficients of steel to oil, $\alpha_o$, air, $\alpha_a$, and gas, $\alpha_g$, are determined as:

$$\alpha_o = 100, \quad \alpha_a = 20, \quad \alpha_g = 25$$

It is noted that the heat transfer coefficients were determined under natural convection conditions. Especially, $\alpha_a$ can change in orders of magnitude if the accumulator exhibits a forced convection [23]. The robustness of the fault detection method developed in this paper is evaluated in Section 5.

As seen, the piston position residual is within $-1$ cm and $-3$ cm during the entire test sequence. In Figure 8, the temperature of two different surface elements can be seen for the

same test. The upper sub-figure in Figure 8 shows the cylinder position and the definitions of the wall element (indicated by the horizontal grid lines). For this test, the piston position is within Wall Element 2 for most of the period. Wall Element 1 is in contact with the oil during the test. Wall Element 8, shown in the lower sub-figure, is only in contact with the gas. For Wall Element 8, the experimental temperature is shown for both the thermocouples attached to the top and bottom.

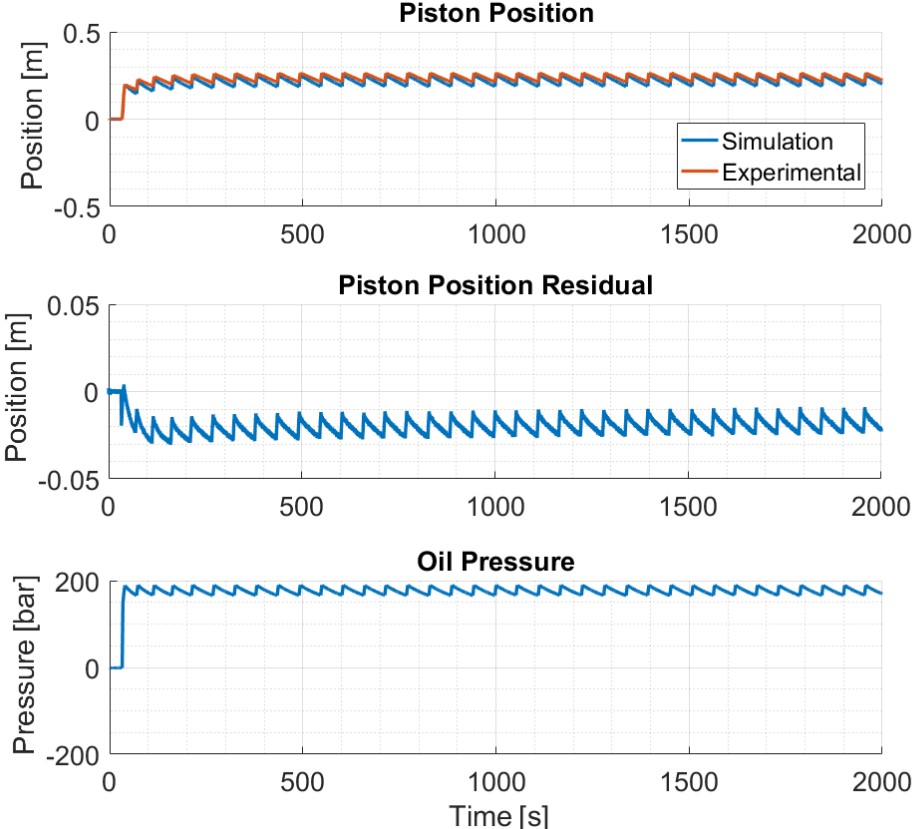

**Figure 7.** Simulated and experimental piston position for a test sequence emulating the operation of a supply accumulator. The position residual is defined as the difference between the simulated and experimental values.

In the lower sub-figure of Figure 8, it is seen that the temperatures at the bottom and the top of the accumulator are not the same. Instead, a temperature difference of up to 2 degrees is seen, indicating that the gas temperature is not uniform when the accumulator is fixed horizontally. The temperature difference may be ascribed to non-uniform gas temperature and different convention conditions over the circumference of the accumulator. As the model assumes a uniform temperature, the simulated temperature falls between the temperatures measured at the accumulator's bottom and top, thus resembling an intermediate temperature.

The results show that the model is partly deviating from the measurements, indicating that some uncertainties influence the model's accuracy. As an example, the assumption of a uniform gas temperature can be seen to cause a steady-state error between the measured and simulated surface temperatures. Therefore, these deviations are expected to cause estimation errors, and the robustness to such model errors is investigated in Section 5. When applied in a wind turbine, the accumulators are rotating in the hub. It is suspected that the gas in a rotating accumulator is mixed to a larger extent than in the fixed configuration, thus leading to a lower temperature gradient in the radial direction compared to the experiments shown in this paper.

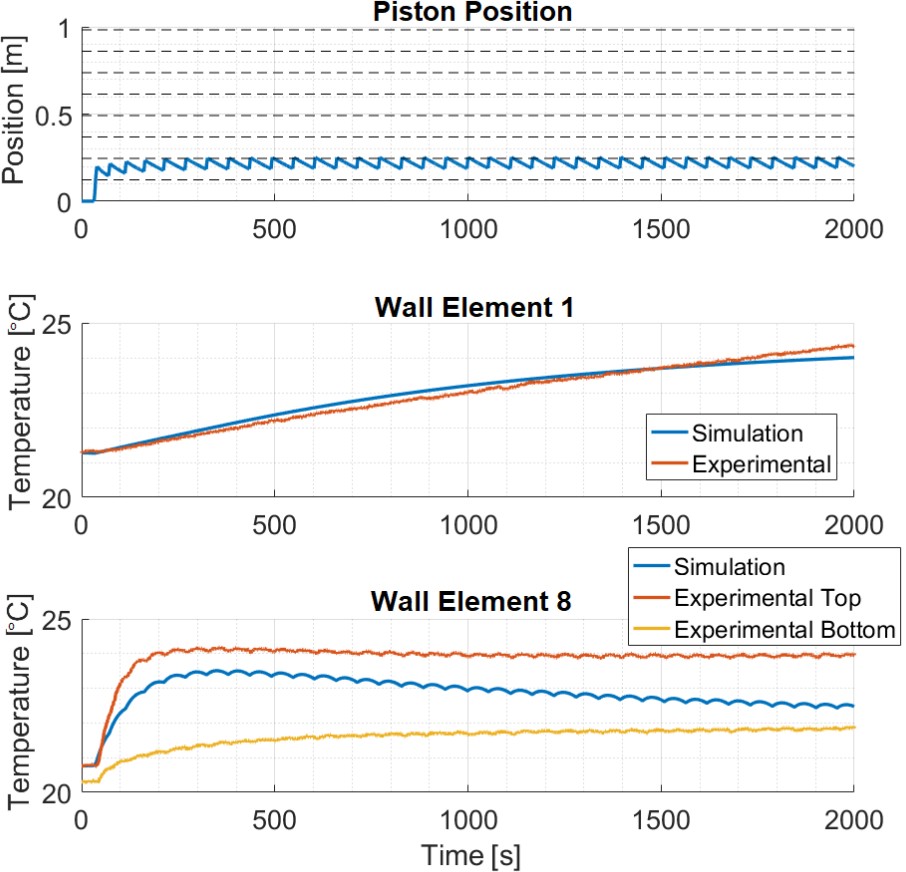

**Figure 8.** Simulated and experimental wall temperature of two wall elements. The upper sub-figure shows the piston position during the test. The dashed lines indicate the wall elements.

## 4. Estimator Design

For estimating changes in the pre-charge pressure of the accumulator, a State Augmented Extended Kalman Filter (SAEKF) was designed based on the model described above. The filter is used to obtain a direct estimate of the pre-charge pressure. In this section, the implementation of the SAEKF is presented based on the work by [24]. For the SAEKF, the model was augmented with an additional state—$n_{N2}$—which represents the number of mol of nitrogen in the accumulator and is a more direct measure of the amount of remaining gas. Furthermore, the number of mol of nitrogen can be used together with the gas temperatures to determine the pre-charge pressure, another standard measure of remaining accumulator gas. The state and input vectors of the SAEKF are, thus:

$$\mathbf{x} = \begin{bmatrix} x_p & \dot{x}_p & T_w^{[1]} & \cdots & T_w^{[8]} & T_e & T_p & T_g & p_g & n_{N2} \end{bmatrix}^T \tag{25}$$
$$\mathbf{u} = \begin{bmatrix} p_o & T_o & T_a \end{bmatrix}$$

The state transition model and observation model can be described as a stochastic process, as shown in Equation (26).

$$\mathbf{x}[k] = \mathbf{f}(\mathbf{x}[k-1], \mathbf{u}[k]) + \mathbf{w}[k] \tag{26}$$
$$\mathbf{y}[k] = \mathbf{h}(\mathbf{x}[k]) + \mathbf{v}[k] \tag{27}$$

where $\mathbf{w} \in \mathbb{R}^m$ is the process noise and $\mathbf{v} \in \mathbb{R}^n$ the measurement noise. Both $\mathbf{w}$ and $\mathbf{v}$ are assumed to be zero mean multivariate Gaussian noises with covariance $\mathbf{Q} \in \mathbb{R}^{m \times m}$ and $\mathbf{R} \in \mathbb{R}^{n \times n}$, respectively. $[k]$ and $[k-1]$ represent the current and previous time step.

The non-linear state transition function $\mathbf{f}(\mathbf{x}, \mathbf{u})$ is shown below:

$$x_p[k] = x_p[k-1] + T_s \dot{x}_p \tag{28}$$

$$\dot{x}_p[k] = \dot{x}_p[k-1] + T_s \frac{(p_o - p_g)A_p - F_{fric}}{m_p} \tag{29}$$

$$T_w^{[i]}[k] = T_w^{[i]}[k-1] + \frac{T_s}{\rho c_{steel}} \left( \frac{\dot{Q}_{aw}^{[i]} + \dot{Q}_{ow}^{[i]} + \dot{Q}_{gw}^{[i]}}{(r_{os}^2 - r_{is}^2)\pi d_x} + \lambda_x \frac{T_o - 2T_w^{[i]} + T_w^{[i+1]}}{d_x^2} \right) \tag{30}$$

$$T_e[k] = T_e[k-1] + T_s \frac{\dot{Q}_{ge} + \dot{Q}_{ae}}{m_e c_{steel}} \tag{31}$$

$$T_p[k] = T_p[k-1] + T_s \frac{\dot{Q}_{gp} + \dot{Q}_{op}}{m_p c_{steel}} \tag{32}$$

$$T_g[k] = T_g[k-1] + T_s \frac{\dot{Q}_s}{c_{N2} \, n_{N2} \, M_{N2}} + \frac{\dot{v} T_g}{c_{N2} \, M_{N2}} \left( \frac{\partial p_g}{\partial T_g} \right) \tag{33}$$

$$p_g[k] = \frac{RT_g}{v} + \frac{B_0 RT_g - A_0 - \frac{C_0}{T_g^2}}{v^2} + \frac{bRT_g - a}{v^3} + \frac{a\alpha}{v^6} + \frac{c\left(1 + \frac{\gamma}{v^2}\right)e^{-\frac{\gamma}{v^2}}}{v^3 T_g^2} \tag{34}$$

$$n_{N2}[k] = n_{N2}[k-1] \tag{35}$$

where $T_w^{[i]}[k]$ is calculated for $i = 1 : 8$ with the last element being $T_w^{[9]} = T_e$. Recall that $v$ and $\dot{v}$ are both a function of piston position, $x_p$, and molar value, $n_{N2}$, as given in Equations (4) and (6).

The output function $\mathbf{h}(\mathbf{x}[k])$ is shown below:

$$\mathbf{h}(\mathbf{x}[k]) = \begin{bmatrix} \mathbf{0}_{9\times2} & \mathbf{I}_9 & \mathbf{0}_{9\times4} \end{bmatrix} \mathbf{x}[k] \tag{36}$$

The filter first calculates a prior state estimate, $\mathbf{x}[k]^-$, by the discrete model shown in Equation (26):

$$\hat{\mathbf{x}}[k]^- = \mathbf{f}(\mathbf{x}[k-1], \mathbf{u}[k]) \tag{37}$$

The prior state error covariance matrix, $\mathbf{P}[k]^-$, and the Kalman gain, $\mathbf{K}[k]$, is then calculated as shown in Equations (38) and (39).

$$\mathbf{P}^-[k] = \mathbf{A}[k]\mathbf{P}^+[k]\mathbf{A}[k]^T + \mathbf{Q} \tag{38}$$

$$\mathbf{K}[k] = \mathbf{P}[k]^- \mathbf{H}[k]^T \left( \mathbf{H}[k]\mathbf{P}^-[k]\mathbf{H}[k]^T + \mathbf{R} \right)^{-1} \tag{39}$$

where $\mathbf{A} \in \mathbb{R}^{m \times m}$ and $\mathbf{H} \in \mathbb{R}^{n \times m}$ are Jacobian matrices defined as:

$$\mathbf{A}[k] = \left. \frac{\partial \mathbf{f}}{\partial \mathbf{x}} \right|_{\hat{\mathbf{x}}[k-1]^+, u[k]} \qquad \mathbf{H}[k] = \left. \frac{\partial \mathbf{h}}{\partial \mathbf{x}} \right|_{\hat{\mathbf{x}}[k]^-}$$

The Kalman gain is then used to calculate the posterior state estimate, $\hat{\mathbf{x}}[k]^+$, and state error covariance matrix, $\mathbf{P}[k]^+$, as shown in Equations (40) and (41):

$$\hat{\mathbf{x}}[k]^+ = \hat{\mathbf{x}}[k]^- + \mathbf{K}[k](\mathbf{z}[k] - \mathbf{h}(\hat{\mathbf{x}}[k]^-)) \tag{40}$$

$$\mathbf{P}[k]^+ = (\mathbf{I} - \mathbf{K}[k]\,\mathbf{H}[k])\mathbf{P}[k]^- \tag{41}$$

where $\mathbf{z}[k]$ is the measurement of sample $k$. The measurements are the wall and end cap temperature measured on top of the accumulator, as shown in Figure 6. The choice of the covariance matrices $\mathbf{Q}$ and $\mathbf{R}$ is important for the performance of the filter, as they have an influence on the dynamic performance. For this paper, $\mathbf{Q}$ and $\mathbf{R}$ were chosen as the diagonal matrices:

$$\mathbf{Q} = \text{diag}\begin{pmatrix} 0.5 & 0.5 & 10 & 10 & 5\ldots 5 & 5000 \end{pmatrix} \tag{42}$$

$$\mathbf{R} = \text{diag}\begin{pmatrix} 0.5 \ldots 0.5 \end{pmatrix} \tag{43}$$

This completes the SAEKF description. In the following section, both the simulation and experimental results are presented, showing the filter's performance.

## 5. Results

The pre-charge charge estimation was conducted for five levels of pre-charge pressure ranging from 145 to 95 bar. To evaluate the estimator in different conditions, the accumulator was run in two different load sequences: the one where the fluid is discharged over a fixed orifice to the tank is denoted as low load, and the one where the discharge oil flow follows a sequence similar to a load cycle in a wind turbine is denoted as high load. The low- and high-load scenarios do not have an explicit effect on the mathematical model due to the assumption of a uniform gas temperature; yet, for the physical system, the gas is mixed to a larger extent when the piston moves faster, thereby affecting the conduction conditions between the gas and accumulator. Selecting high- and low-load conditions, therefore, allows for testing robustness due to these non-modeled effects. The test bench was located at an ambient temperature of 21 °C, and the fluid temperature was controlled in the reservoir at 45 °C.

The duration of the load sequences was chosen to be 2000 s, which is long enough for capturing the main thermal dynamics. The input to the estimator SAEKF $\mathbf{u} = [p_a, T_o, T_a]^T$ was given from experimental data. The temperature of the wall and accumulator were used as measurements, $\mathbf{z} = \left[ T_w^{[1]}, \ldots, T_w^{[8]}, T_e \right]^T$, as shown in Equation (25).

The state vector was initialized as the first entrance of $\mathbf{z}$ for the measured states. The piston position and velocity were initialized as zero. The piston and gas temperature, $T_p$ and $T_g$, were initialized as the mean of all the wall elements' temperatures. The gas pressure, $p_g$, was initialized as the measured pre-charge pressure, and the molar value, $n_{N2}$, was calculated from the measured pre-charge pressure from Equation (3) when the gas temperature has settled to the ambient temperature. The pre-charge pressure was read using a manometer. The low-load sequence was used for the experimental result. The estimated and measured piston positions are shown together with the estimated and measured molar values in Figure 9.

As seen, the estimated piston position and molar value converged after a transient period of 600 s to a value just below the target. Comparing the target to the average of the estimated molar value from 1800–2000 s yielded a steady-state error of −3%. The results of the high-load test sequence are shown in Figure 10.

The estimation at the high-load scenario showed a similar transient behavior as for the low-load sequence. The discrepancy between the estimated and target molar value was larger than for the low-load sequence, valued at −8%. In general, for all the tested pre-charge levels and load sequences, the estimated values converged after a similar transient period, as shown in Figures 9 and 10. The presented estimation technique, therefore, clearly solves the drifting problem presented in the literature. The steady-state results of the experiments are shown in Figure 11. The first axis shows the measured pre-charge pressure $p_{0,meas}$, and the second axis shows the estimated value $p_{0,est}$. The error bars show the first standard deviation of the values used for determining the average estimated pre-charge pressure. The standard deviation was small in all experimental results.

The estimation was seen to fall above and below the target value with a maximum error of 10% in the low-load 95 bar experiment. The error was ascribed to the model uncertainties caused by the heat transfer coefficients and uniform gas assumption, where the model yielded a steady-state error, as indicated in Figure 8. Despite the error, the estimated values generally followed the tendency of the measured pre-charge pressure. While the estimation from the low-load sequence at 135 bar pre-charge cannot be isolated from the

high-load sequence at 145 bar, all other levels can be isolated irrespective of the load sequence. Generally, the high-load sequence reduced the estimated value.

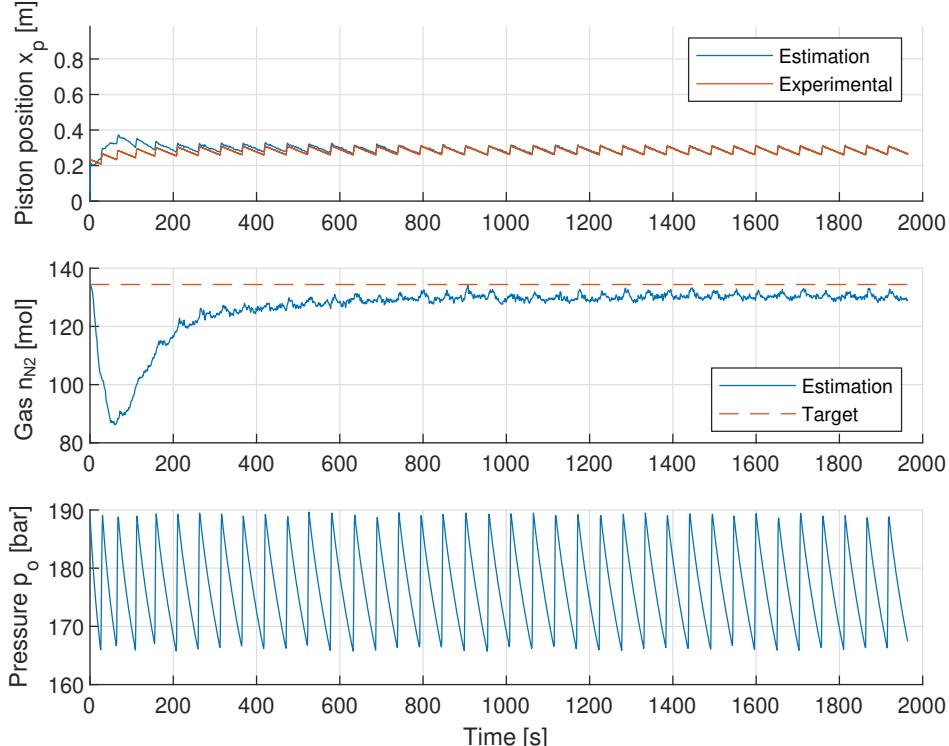

**Figure 9.** Estimated and experimentally measured piston position and molar value of gas for the low-load test sequence. The accumulator is pre-charged to 125 bar/134 mol.

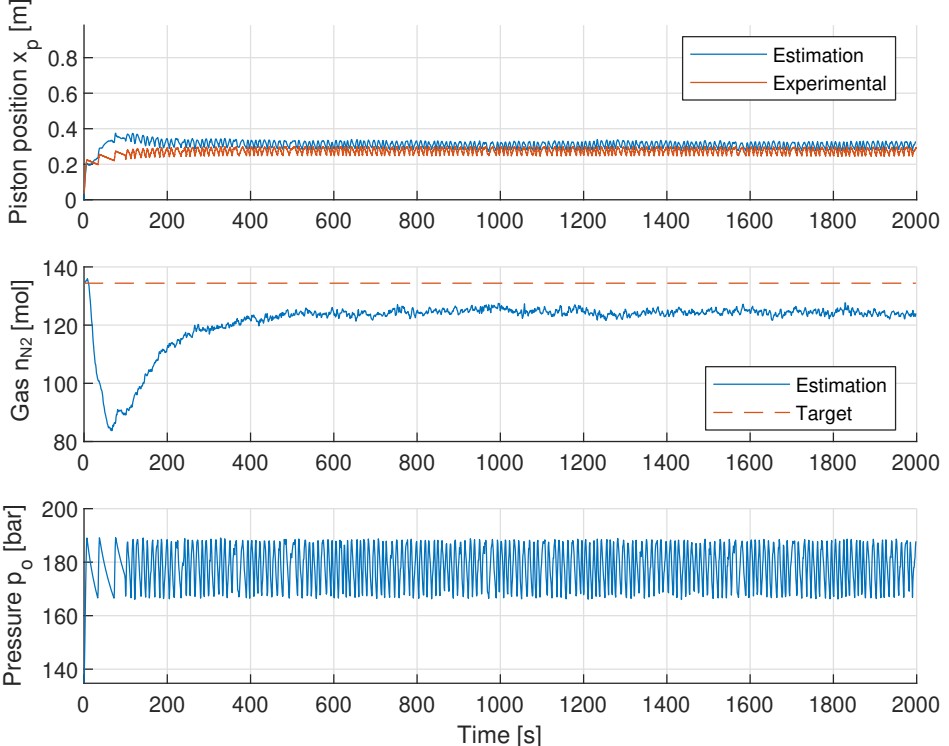

**Figure 10.** Estimated and experimentally measured piston position and estimated molar value of gas for the high-load test sequence. The accumulator is pre-charged to 125 bar/134 mol.

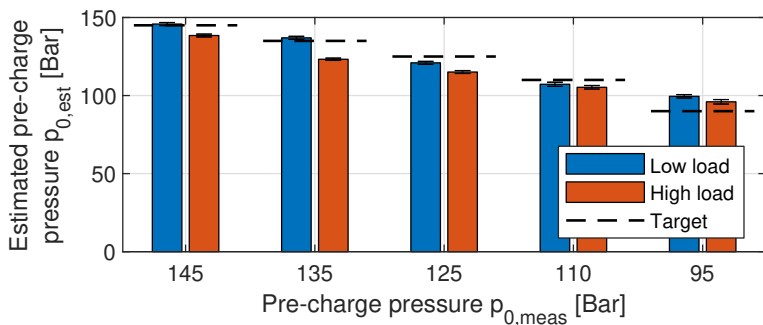

**Figure 11.** Estimated pre-charge pressures at selected levels for the low-load and high-load test sequences.

To further show that the estimation converged, a test where the estimated molar value was initialized at different values was conducted. The estimated molar values were initialized at $\pm 20\%$ of the measured value. The results for the low-load sequence and a pre-charge pressure of 125 bar are shown in Figure 12.

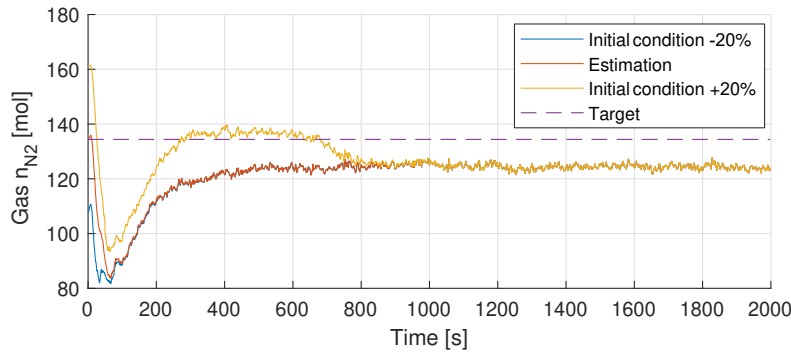

**Figure 12.** Estimated and experimental molar value for the low-load test sequence at different initial conditions. The accumulator is pre-charged to 125 bar/134 mol.

As seen, the estimation converged towards the same value despite the uncertainty of the actual pre-charge level of the accumulator.

The robustness of the method to changing convention conditions was tested by forcing air axially along the accumulator using a fan. The wind speed was measured as 5 m/s, which is more than what can be expected in the hub of a wind turbine. The results are shown in Figure 13.

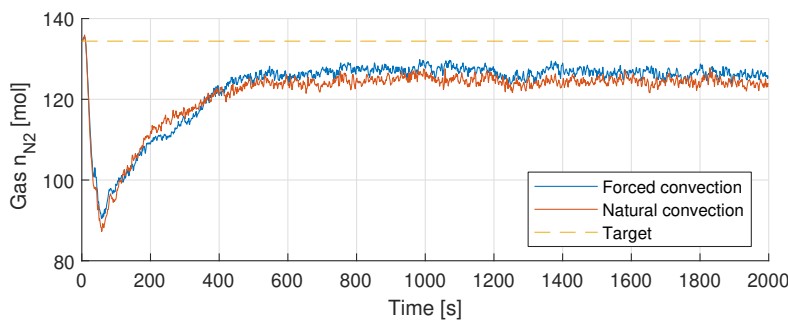

**Figure 13.** Estimated and experimental molar value for the low-load test sequence under natural and forced convection conditions. The accumulator is pre-charged to 125 bar/134 mol.

The transient period was similar for the two conditions, while the converged value was slightly higher for forced convection. The difference in the estimated value is limited to a few percent, and thus, the method is robust to changing convection conditions. It

is noted that the surface temperatures were reduced below 1 K between the forced and natural convection conditions.

Conclusively, the estimated values show a difference between the tests with different pre-charge pressures. Even though the estimated pre-charge pressure shows an offset, the estimate may still be compared to an estimated value of a "fault-free" accumulator, and in that way, it can detect a potential change in pre-charge pressure. Here, it should also be considered that a high load, i.e., high flow in and out of the accumulator, influences the estimate, however to a significantly lower extent.

## 6. Conclusions

This paper investigated gas leakage detection by the estimation of the pre-charge pressure of a hydraulic piston accumulator using a State Augmented Extended Kalman Filter, which used temperature measurements on the accumulator surface as the inputs. Contrary to previous methods, this approach enabled the ability to isolate gas leakage to a single accumulator out of a bank of accumulators. The SAEKF was based on a model of the accumulator, which estimates the number of moles of nitrogen gas in the accumulator, which together with the gas temperature, was used to determine the pre-charge pressure. The SAEKF was tested on an experimental setup using low and high accumulator flow, i.e., varying load, and for five different pre-charge pressures. Based on the experiments, it was found that the estimated pre-charge pressure drift, which has been reported in previous works, was solved. Furthermore, it was found that, for the considered experiments, the estimation was associated with an offset up to 10%, which partly was attributed to the uncertainties with the used heat transfer coefficients and model assumption of uniform gas temperatures. Despite the offsets in the estimated pre-charge pressure, the SAEKF was shown to settle to constant values for a given pre-charge pressure. The method can, therefore, be used to detect changes in the pre-charge pressure by comparing different measurements over time to detect any change in the estimated pre-charge pressure in spite of this offset.

**Author Contributions:** Methodology, M.F.A. and J.L.; software, J.L.; validation, J.L.; writing—original draft preparation, M.F.A. and J.L.; writing—review and editing, J.L., N.S. and H.C.P.; supervision, J.L., N.S. and H.C.P.; funding acquisition, H.C.P. All authors have read and agreed to the published version of the manuscript.

**Funding:** This research is funded by Innovation Fund Denmark: 8053-00039.

**Conflicts of Interest:** The authors declare no conflict of interest.

## Abbreviations

The following abbreviations are used in this manuscript:

| | |
|---|---|
| EKF | Extended Kalman Filter |
| SAEKF | State Augmented Extended Kalman Filter |
| BWR | Benedict–Webb–Rubin |
| MW | Mega-Watt |

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
