# Peer review of "Pre-Charge Pressure Estimation of a Hydraulic Accumulator Using Surface Temperature Measurements"

_2674-032X, doi:10.3390/wind2040041_

Round 1

Reviewer 1 Report

1) In the introduction was mentioned that : "The main failure mode of a gas-charged piston accumulator is loss of pre-charge pressure, i.e. gas leakage ...". Where was gas leakage taken into account in the physical model of the accumulator? Even if this phenomenon is indirectly accounted for through the gas pressure drop, it still exists and the system is no longer a closed system. Can you elaborate on this subject? If leakage is present the whole model below does no longer hold.

2) Eq. 4 looks quite odd in terms of 1st Law. The heat flow in left hand side should not bear a minus sign as long as you do not presume a certain direction of the heat flow (i.e., from gas to walls or from walls to gas). How do you know what is the order relation between the temperature of the wall compared to the temperature of the gas? Secondly, if v is the specific molar volume [m3/mol] what is MN2 ? Molar density ? How do you define it? From my point of view this molar density reads as mol/m3 which means the reciprocal of v. If so, Eq. 4 is no longer dimensionally correct. Instead, it is more likely the molar mass [kg/mol] of nitrogen.

3) Fig. 4 your refer to on p. 6 has disappeared.

4) Eq. 18: what are taking the derivative to? The product between conductivity and the derivative of temperature all together or separately. If former, then a bracket should do the job.

5) Line 180: you are mentioning that fluid is purged into a tank through a valve. It means the system is not closed!!!! And this is the explanation for the oscillations presented in Fig. 7 in the position of the piston. At least, this how I can explain these variations of the position of the piston.

6) How do you handle the molar volume v required in Eq. 30 from the equation of state 31) which is a strong non-linear equation?

7) It is not clear for me how do you model the low and high loads. What is the impact of these loads on the physical and mathematical model (Eqs. 1-22 or Eqs. 23-32).

8) What do you mean by "manually measured pre-charge pressure"?

9) The estimated value for the number of moles mmol in Figs. 10 and 12 is quite far from the target. More than 10%? How do you explain this unless is a consequence of using a closed system?

Author Response

Dear Reviewer,

We are thankful for the insightful and detailed comments and suggestions. It is always fruitful to read and consider such thorough reviews. Below you may find a point-to-point description of our changes, corrections, and updates based on the comments. The updated manuscript changes are marked with red.

1) In the introduction was mentioned that : "The main failure mode of a gas-charged piston accumulator is loss of pre-charge pressure, i.e. gas leakage ...". Where was gas leakage taken into account in the physical model of the accumulator? Even if this phenomenon is indirectly accounted for through the gas pressure drop, it still exists and the system is no longer a closed system. Can you elaborate on this subject? If leakage is present the whole model below does no longer hold.

The reviewer points to an important mistake since we have not presented our considerations on the gas leakage fault in the model. The gas leakage occurs either in discrete events or very slowly developing over weeks or months. Therefore, it is suitable to assume a constant amount of gas within the time frame of detection used in this paper. This crucial point is now elaborated on from line 134 in the new manuscript.

2) Eq. 4 looks quite odd in terms of 1st Law. The heat flow on the left-hand side should not bear a minus sign as long as you do not presume a certain direction of the heat flow (i.e., from gas to walls or from walls to gas). How do you know what is the order relation between the temperature of the wall compared to the temperature of the gas? 

We acknowledge that it has not been evident in which direction the heat flows were defined due to the absence of fig. 4. This is now corrected in the new manuscript. dQ_s is defined such that it describes heat flow from the gas to the interfacing thermal elements, such as the wall elements, piston, and end cap as shown in Eq. 22. Thus, a positive dQ_s should cause the gas temperature gradient to lower, hence, the minus. Line 141 in the update has been changed to accommodate this point.

Secondly, if v is the specific molar volume [m3/mol] what is MN2 ? Molar density? How do you define it? From my point of view this molar density reads as mol/m3 which means the reciprocal of v. If so, Eq. 4 is no longer dimensionally correct. Instead, it is more likely the molar mass [kg/mol] of nitrogen.

The reviewer correctly points out that M_N2 is not the molar density but the molar mass. This is now corrected in line 143 in the new manuscript.

3) Fig. 4 your refer to on p. 6 has disappeared.

Unfortunately, this error occurred during pdf translation and we did not thoroughly check the results. The figure is now present in the pdf version of the manuscript and apologizes for the inconvenience.

4) Eq. 18: what are taking the derivative to? The product between conductivity and the derivative of temperature all together or separately. If former, then a bracket should do the job.

We mistakenly missed a bracket in both Eq. (20) and the line equation in the line just below the expression. This is not corrected in the updated manuscript.

5) Line 180: you are mentioning that fluid is purged into a tank through a valve. It means the system is not closed!!!! And this is the explanation for the oscillations presented in Fig. 7 in the position of the piston. At least, this how I can explain these variations of the position of the piston.

We acknowledge that the test sequence has not been fully described. The test sequence used to generate the experimental data is chosen to simulate typical operation for an accumulator placed in the supply unit of a hydraulic pitch control system which is also used in previous works (see citation in the manuscript). Here the pressure level is controlled between two thresholds chosen as 165 bar and 190 bar. A constant flow supply is turned on when the pressure is below the lower threshold. The supply is turned off when the pressure rises to the upper threshold pressure. This is what causes the saw-tooth-like pressure changes. The fluid flow from the accumulator is controlled in closed loop to simulate the flow demand from a pitch system where the flow varies due to the volatility of the pitch activity. We have updated the description with more details in the new manuscript from line 170.

6) How do you handle the molar volume v required in Eq. 30 from the equation of state 31) which is a strong non-linear equation?

This has not been clear in the original manuscript. The equations have been changed to describe the dependency on the molar amount of nitrogen instead of the mass which also increases the readability of the accumulator model. The changes have been implemented by adding Eq. (4)+(6) and changing lines 141 + 218, Eq. (25), Eq. (32)+(34) in the new manuscript.

7) It is not clear for me how do you model the low and high loads. What is the impact of these loads on the physical and mathematical model (Eqs. 1-22 or Eqs. 23-32).

We acknowledge that this has not been described sufficiently and have added the following in the updated manuscript from line 228: The low and high load scenarios do not have an explicit effect on the mathematical model due to the assumption of homogeneous gas temperature, yet, for the physical system the gas is mixed to a larger extent when the piston moves faster. Thereby, affecting the conduction conditions between the gas and accumulator. Selecting high and low load conditions, therefore, allows for testing robustness due to these non-modeled effects.

8) What do you mean by "manually measured pre-charge pressure"?

The term "manual" confuses the message and is deleted in the new manuscript. The pressure is read from an analog manometer, as also described in the updated manuscript from line 240.

9) The estimated value for the number of moles mmol in Figs. 10 and 12 are quite far from the target. More than 10%? How do you explain this unless is a consequence of using a closed system?

We are accounting this error to the simplified way the thermal behavior of the gas and accumulator relation is modelled. A more detailed model using CFD results could reveal that the gas is non-uniform in temperature as indicated in Figure 8. The figure reveals a steady state error between the simulation and measurement which causes a similar steady-state error in the estimated amount of gas. We have clarified this point in the updated manuscript from lines 207 and 262. 

Reviewer 2 Report

The presented article presents the extension of the mathematical model used to identify the pre-charge pressure status of hydraulic accumulators realized by measuring the external temperature. The work contains an interesting and, more importantly, validated by real tests model, in which the level of usefulness seems to be very high. The presented model can be used to diagnose the state of pre-charge of hydraulic piston accumulators used in various machines and systems, not only in wind turbines. However, the article has a significant deficiency at the editorial level, in the form of missing figures 5 and 6. Moreover, the article does not have a summary and ends with a presentation of the research results. In addition, a typing error was noticed in Figure 8 (buttom -> bottom).

Author Response

Dear Reviewer,

We are thankful for the insightful comments and suggestions. Below you may find a point-to-point description of our changes, corrections, and updates based on the comments. The updated manuscript changes are marked with red.

1)The presented article presents the extension of the mathematical model used to identify the pre-charge pressure status of hydraulic accumulators realized by measuring the external temperature. The work contains an interesting and, more importantly, validated by real tests model, in which the level of usefulness seems to be very high. The presented model can be used to diagnose the state of pre-charge of hydraulic piston accumulators used in various machines and systems, not only in wind turbines. However, the article has a significant deficiency at the editorial level, in the form of missing figures 5 and 6. Moreover, the article does not have a summary and ends with a presentation of the research results. In addition, a typing error was noticed in Figure 8 (buttom -> bottom).

We thank the reviewer for the comments and we apologize for the editoral mistakes. The figures were not translated well when the manuscript was converted to pdf. Figures 4,5,6 are now inserted correctly. A conclusion is also added to the updated manuscript presenting the research result. The typing error in the legend of Figure 8 is also corrected in the new manuscript.

Reviewer 3 Report

It is recommended that the author make changes based on the following comments:

1.The necessity of innovative research in this paper needs to be further reflected in the abstract and introduction.

2.Literature sources for many of the formulas in the text need to be given.

3.The format of formula (33) needs to be unified with other formulas.

4.The paper lacks much of the original data.

5.This paper lacks field experiments and analysis.

6.The author needs to add a conclusion section. Conclusions allow authors to have the final say on the questions raised in their papers, demonstrate the importance of their research, and push readers to have new perspectives on the research. Conclusions can go beyond the limitations of research, allowing authors to consider broader questions, make new connections, and articulate the implications of new findings. Thus, the conclusion will help readers understand why all the author's analysis and information is important to them after they put down the paper.

This paper can be accepted after minor revision by the above suggestions.

Author Response

Dear Reviewer,

We are thankful for the insightful and detailed comments and suggestions. It is always fruitful to read and consider such thorough reviews. Below you may find a point-to-point description of our changes, corrections, and updates based on the comments. The updated manuscript changes are marked with red.

1.The necessity of innovative research in this paper needs to be further reflected in the abstract and introduction.

We have updated the with more thorough descriptions of the innovations in the updated manuscript in line 7 of the abstract and starting from line 94 and line 131 in the introduction. We hope, that the innovation of the developed method is now sufficiently clear.

2.Literature sources for many of the formulas in the text need to be given.

We have not updated with citation of equations (3), (20). A source for Equations (25)-(27) describing the EKF has now been commonly cited in line 221 of the updated manuscript.

3.The format of formula (33) needs to be unified with other formulas.

The format has now been corrected for what is now equations (36) in the updated manuscript.

4.The paper lacks much of the original data.

The parameter values of the model presented are now collected in table 1 of the updated manuscript.

5.This paper lacks field experiments and analysis.

The field analysis is considered in the selection of load flow conditions, yet, we agree that this has not been fully elaborated on in the original manuscript. Further details are now available in the updated manuscript starting from line 235.

Unfortunately, we do not have an operational wind turbine available for testing the detection method in a field environment. This would be an obvious next step for furthering the usability of the method. We hope that the reviewer can agree that the laboratory test performed in the manuscript provides sufficient contribution for publication.

6.The author needs to add a conclusion section. Conclusions allow authors to have the final say on the questions raised in their papers, demonstrate the importance of their research, and push readers to have new perspectives on the research. Conclusions can go beyond the limitations of research, allowing authors to consider broader questions, make new connections, and articulate the implications of new findings. Thus, the conclusion will help readers understand why all the author's analysis and information is important to them after they put down the paper.

The conclusion has been omitted in the submitted manuscript entirely by mistake. We have now added a conclusion to the manuscript and apologize for the inconvenience.

Reviewer 4 Report

This is a nice and consistent paper about a relevant topic. The research approach is adequate. The reviewer asks only for a few modifications/amendments:

Conclusions are missing: Are suggestions for an improvement of the model? Is the achievable prediction quality for the gas content sufficient? Provide numbers of requested accuracy for knowing the actual pre-charge pressure or mol number, respectively, and the achievable accuracy.

Results show a negligible role of the heat exchange coefficient from the wall to the air. Can a plausible explanation be given for this?

The nitrogen volume is handled with a uniform temperature. This is definitely valid for the thermos-elastic effect (temperature change due to compression/expansion) but heat exchange with the wall might cause non-uniform gas temperature. Even though the model seems adequate for the envisaged use case it might have insufficient accuracy in other cases. This should be discussed. In this context it is helpful to explain the use case more in detail, for instance, by showing graphs of the input (u) quantities which were used.

There are a few minor improvements:

Page 2, line 49: “100 bar and 50 could be isolated” from each other. Find a better wording and discuss, if this is sufficient for the wind-mill application.

Page 3, line 100: “5 minutes constant fluid pressure charging”. Does it mean that the pressure was raised very quickly and kept constant(by some pressure control) for 5 minutes?

Page 5, line 191 and formula (11): T is the temperature and not directly the energy balance, as stated. Be more precise.

Page 6: Dot missing after Figure 4.

Page 8, line 194: The different temperatures on the top and bottom (Figure 8) are explained resulting from a non-uniform gas pressure. In an industrial related work of the reviewer for the explanation of temperature induced eccentricities of paper mill rolls, the reasons was found being temperature and speed differences of the ambient air which can be significant, even indoor. This could be mentioned as a potential cause too.

Page 11, line 231: “position is shown” -> “positions are shown”

Author Response

Dear Reviewer,

We are thankful for the insightful and detailed comments and suggestions. It is always fruitful to read and consider such thorough reviews. Below you may find a point-to-point description of our changes, corrections, and updates based on the comments. The updated manuscript changes are marked with red.

Conclusions are missing: Are suggestions for an improvement of the model? Is the achievable prediction quality for the gas content sufficient? Provide numbers of requested accuracy for knowing the actual pre-charge pressure or mol number, respectively, and the achievable accuracy.

The conclusion has mistakenly been omitted in the submitted manuscript. We have now added a conclusion to the manuscript and apologize for the inconvenience.

Results show a negligible role of the heat exchange coefficient from the wall to the air. Can a plausible explanation be given for this?

We have investigated and compared surface temperature measurements for the forced and natural convection cases. The surface temperatures are lowered consistently by below 1 degC for the forced convection case and we attribute this to the small change in estimation. This has not also been noted in the updated manuscript in line 287.

The nitrogen volume is handled with a uniform temperature. This is definitely valid for the thermos-elastic effect (temperature change due to compression/expansion) but heat exchange with the wall might cause non-uniform gas temperature. Even though the model seems adequate for the envisaged use case it might have insufficient accuracy in other cases. This should be discussed. In this context, it is helpful to explain the use case more in detail, for instance, by showing graphs of the input (u) quantities which were used.

We would like to thank the reviewer for the insightful comments. The input oil pressure p_o is seen for each operating flow condition in figure (7), (9) and (10), yet, we have not clearly specified that the oil temperature and ambient temperatures are kept near constant for the laboratory setup. This has now been elaborated in line 240 of the updated manuscript.

There are a few minor improvements:

Page 2, line 49: “100 bar and 50 could be isolated” from each other. Find a better wording and discuss, if this is sufficient for the wind-mill application.

We have updated the wording and inserted further details in the updated manuscript from line 51 and line 92.

Page 3, line 100: “5 minutes constant fluid pressure charging”. Does it mean that the pressure was raised very quickly and kept constant(by some pressure control) for 5 minutes?

Yes, that is correct. We have changed the sentence in line 101 to hopefully highlight this point.

Page 5, line 191 and formula (11): T is the temperature and not directly the energy balance, as stated. Be more precise.

We have updated the description and added a reference citation for the equation in line 150 of the updated manuscript.

Page 6: Dot missing after Figure 4.

This is now updated in the new manuscript.

Page 8, line 194: The different temperatures on the top and bottom (Figure 8) are explained resulting from a non-uniform gas pressure. In an industrial related work of the reviewer for the explanation of temperature induced eccentricities of paper mill rolls, the reasons was found being temperature and speed differences of the ambient air which can be significant, even indoor. This could be mentioned as a potential cause too.

This is a very relevant point and has been incorporated in the updated manuscript starting on line 211

Page 11, line 231: “position is shown” -> “positions are shown”

The grammatical error has been corrected in the updated manuscript.

Round 2

Reviewer 1 Report

Accept in present form.